# Comorbidities and Complications in People Admitted for Leprosy in Spain, 1997–2021

**DOI:** 10.3390/life14050586

**Published:** 2024-05-02

**Authors:** Blanca Figueres-Pesudo, Héctor Pinargote-Celorio, Isabel Belinchón-Romero, José-Manuel Ramos-Rincón

**Affiliations:** 1Internal Medicine Service, General University Hospital Dr. Balmis, Alicante Health and Biomedical Research Institute (ISABIAL), 03010 Alicante, Spain; bfiguerespesudo@gmail.com (B.F.-P.); jose.ramosr@umh.es (J.-M.R.-R.); 2Infectious Disease Unit, Internal Medicine Service, General University Hospital Dr. Balmis, Alicante Health and Biomedical Research Institute (ISABIAL), 03010 Alicante, Spain; hecdtorpinargote@gmail.com; 3Dermatology Service, General University Hospital Dr. Balmis, Alicante Health and Biomedical Research Institute (ISABIAL), 03010 Alicante, Spain; 4Clinical Medicine Department, University Miguel Hernández, 03550 Alicante, Spain

**Keywords:** leprosy, hospitalizations, epidemiology, Spain, comorbidity

## Abstract

This study aims to describe the epidemiological and clinical characteristics and trends of these admissions in Spain. This retrospective study drew data from the Hospital Discharge Records Database of the Spanish National Health System. We used the diagnostic codes for leprosy from the International Classification of Diseases, ninth and tenth revisions, to retrieve leprosy admissions from 1997 to 2021. There were 1387 hospitalizations for leprosy The number of annual cases decreased gradually, from 341 cases in 1997–2001 to 232 in 2017–2021 (*p* < 0.001). Patients’ median age increased, from 65 years in 1997–2001 to 76 years in 2017–2021 (*p* < 0.001), as did the prevalence of some comorbidities, such as hypertension (15% in 1997–2001 to 27.6% in 2017–2021; *p* < 0.001). The mortality rate (6%) and the frequency of leprosy complications remained stable. After Spain (79.1%), the most common country of origin was Paraguay (4.4%). Admissions decreased significantly in Andalusia, from 42% in 1997–2001 to 10.8% in 2017–2021 (*p* < 0.001), and in the Canary Islands, from 7.9% in 1997–2001 to 2.6% in 2017–2021 (*p* = 0.001), whereas they increased in Madrid, from 5.9% in 1997–2001 to 12.1% in 2017–2021 (*p* = 0.005). Overall, leprosy admissions in Spain have declined, even in the regions with the highest prevalence. Patients admitted for leprosy have become older and sicker.

## 1. Introduction

Leprosy is an infection caused by Mycobacterium leprae. It mainly affects the skin and peripheral nerves, is responsible for skin and secondary bone and neurological complications and can be associated with marked functional limitations [1,2]. The last decades have seen a decrease in the incidence of leprosy worldwide, from an estimated 641,091 cases in 2000 to 140,594 in 2021 [3,4,5]. In 2021, most cases occurred in resource-poor countries, with 66.5% of reported cases concentrated in South-East Asia, followed by 15.1% in the African Region, 14.1% in the Americas, 2.6% in the Eastern Mediterranean Region, and 1.8% in the Pacific [3].

In Spain, the leprosy burden has also decreased steadily since the second half of the twentieth century (annual decrease in leprosy incidence 1.6%) [6]. At the same time, immigration in Spain increased fivefold between the turn of the 21st century and 2023, with many migrants coming from countries where leprosy is a relevant public health concern [7,8]. In fact, most newly reported cases of leprosy in Spain are foreign nationals from countries where leprosy is endemic, such as Brazil [5,9,10,11,12].

The State Leprosy Registry in Spain started under the management of the National Epidemiology Center in 1992. This register is based on a file of cases, and a final annual report is published every year in the Weekly Epidemiological Bulletin [13,14]. In Spain, leprosy should be considered among the differential diagnoses in patients with cutaneous and neurological signs and symptoms, particularly in migrant populations, as most immigrants living in Spain are from Central and South America, where leprosy remains a relevant concern [7,8]. This is also the case in the USA, where cases of imported leprosy have been described in immigrants from Central and South America [15].

Diabetes mellitus, hypertension, and dyslipidemia are also global health challenges. Areas with high prevalence of diabetes mellitus also present a higher prevalence of hypertension and dyslipidemia. These diseases can coexist in the same patient [16,17], and they also interact with leprosy. However, comorbidities in leprosy patients have not been analyzed in the medical literature [16,17].

The spectrum of the disease, as well as the trends over time in hospital admissions for leprosy, have not been examined to date in Spain, a non-endemic country that has received large migratory flows from endemic regions in Latin America and Africa over the past two decades. The aim of this study is to analyze the evolution of hospital admissions as well as comorbidities and complications in people admitted for leprosy in Spain from 1997 to 2021.

## 2. Materials and Methods

### 2.1. Design and Data Source

This observational, cross-sectional study included patients admitted to Spanish hospitals for leprosy from 1 January 1997 to 31 December 2021. Data were drawn from the hospital discharge records database, also known as the minimum basic data set, which is maintained by the Spanish Ministry of Health. This database collects information on all patients discharged from public hospitals and clinics throughout the country [17,18]. The dataset covers information on sex, age, dates of admission and discharge, length of hospital stay, region (autonomous community) where each hospital is located, up to 20 clinical diagnoses during each episode of hospitalization (on admission and/or during hospital stay), and the circumstances of hospital discharge (e.g., voluntary discharge, transfer to another center, death). 

All cases with leprosy-related diagnoses retrieved from the minimum basic data set, maintained by the Spanish Ministry of Health, were included in the study. No cases were excluded; however, in some instances, certain variables were not recorded, precluding the analysis of specific conditions.

The Spanish Ministry of Health conducts regular audits to assess the accuracy of the data, which can be accessed upon request and are provided following anonymization. Previous studies of this register have been conducted for other conditions, including infectious diseases, and its value for estimating the current burden and time trends has been recognized [19,20].

### 2.2. Variables

Cases were identified according to the leprosy-related diagnostic codes in the International Classification of Diseases, 9th revision, Clinical Modification (ICD-9-CM) from 1997 to 2015 and following ICD-10-CM from 2016 to 2021. Specifically, the ICD-9-CM code was 030 (leprosy), which includes subtypes 030.0 (lepromatous), 030.1 (tuberculoid), 030.2 (indeterminate), 030.3 (doubtful), 030.8 (other forms of leprosy), and 030.9 (unspecified leprosy). The ICD10-CM codes were A30 (leprosy and Hansen’s disease), which includes subtypes A30.0 (indeterminate leprosy), A30.1 (tuberculoid), A30.2 (borderline tuberculoid), A30.3 (borderline), A30.4 (borderline lepromatous), A30.5 (lepromatous leprosy type LL), A30.8 (other forms of leprosy), A30. 9 (unspecified leprosy); and B92 (sequelae of leprosy). The diagnostic codes could appear in any rank on the list of diagnoses for each admission episode.

We collected data on age, sex, length of hospital stay (hospitalizations exceeding 90 days were excluded from this analysis), nationality (only available since 2016), type of leprosy according to ICD-9-CM and ICD-10-CM, leprosy complications (osteomyelitis, cellulitis, ulcerations, and neuropathy), specific comorbidities (including arterial hypertension, dyslipidemia, diabetes mellitus, chronic lung disease, heart failure, cardiac arrhythmia, obesity, anemia, chronic kidney disease, HIV, hepatitis B and hepatitis C virus), and micro-organism responsible for bacterial superinfection.

### 2.3. Statistical Analysis

Categorical variables are presented as absolute and relative values, and continuous variables as mean and standard deviation (SD) or medians and interquartile range (IQR), depending on the normality of their distribution, as determined using the Kolmogorov-Smirnov test. To analyze trends, the 25-year study period was regrouped into 5 five-year periods (1997–2001, 2002–2006, 2007–2011, 2012–2016, and 2017–2021). The linear association between categorical variables was assessed using the chi-squared test in 2 × 5 tables, while continuous variables were compared using the non-parametric Kruskal–Wallis test. All tests were two-tailed, and *p*-values of less than 0.05 were considered statistically significant. Statistical analyses were performed using the IBM SPSS for Windows v25.0 package (IBM Corp., Armonk, NY, USA).

### 2.4. Ethical Aspects

The study protocol was approved by the Clinical Research Ethics Committee of the Alicante General University Hospital (Alicante, Spain) (ref. CEIm: PI2021-119). To ensure patient anonymity, the Spanish Ministry of Health provided the dataset after removing all potential patient identifiers.

## 3. Results

From 1997 to 2021, 1387 hospitalizations for leprosy were recorded, of which 807 (58.2%) were in males and 580 (41.8%) in females. The number of annual admissions decreased from 69 in 1997 to 50 in 2021 (Figure 1), and the rate per 100,000 admissions fell from 2.30 to 1.44 in the same period. The number of annual admissions in 2019 was 43, which decreased to 36 in 2020 (the first year of the SARS-CoV-2 pandemic), and then slightly increased to 50 in 2021 (the second year of the SARS-CoV-2 pandemic. 

### 3.1. Clinical and Epidemiological Characteristics

Table 1 shows the epidemiological characteristics of the 1387 patients admitted for leprosy. Their median age was 68 years (IQR 55–77), with patients younger than 15 years accounting for just 0.3% of all leprosy admissions in the 25 years of study. The most common leprosy-related diagnoses were unspecified leprosy (*n* = 760; 54.8%) and lepromatous leprosy (*n* = 428; 30.9%). Leprosy sequelae (only available since 2016) were described in 3.9% of the cases. The most common comorbidities were hypertension (*n* = 296; 21.3%), diabetes mellitus (*n* = 253; 18.2%), anemia (*n* = 156; 11.2%) and dyslipidemia (*n* = 147; 10.6). The median hospital stay was 8 days (IQR 4–16 days), and 6% of those admitted died. 

The main comorbidities in included patients were hypertension (21.3%), diabetes mellitus (18.2%), anemia (11.2%), and dyslipidemia (10.6%). The most common co-infections were hepatitis C (4.1%) and HIV (2.8%). Regarding complications, 12.9% showed skin ulceration, 6.4% neuropathy, and 5.0% cellulitis. Among the patients with a bacterial superinfection, the main micro-organisms responsible were *Staphylococcus* spp. (3.8%) followed by *Pseudomonas* spp. (2.5%).

Patients’ nationality was available from 2016. After Spain (79.1%), the most common countries of origin were Paraguay (4.4%), Colombia (2.4%), and Senegal (2.4%) (Table 2). Geographically, admissions were concentrated in Andalusia (26.2%), the Valencian Community (23.8%), Catalonia (13.6%), Madrid (9.1%), and the Canary Islands (6.3%) (Table 3).

### 3.2. Trends by Five-Year Study Periods

Over the 25-year study period (Table 4), the median age of leprosy patients increased, from 65 years in 1997–2001 to 76 years in 2017–2021 (*p* < 0.001). There was also a decrease in the percentage of males and an increase in the percentage of females (*p* < 0.001).

Median hospital stay decreased over the study period (*p* = 0.002), while mortality rates remained stable. The number of patients diagnosed with unspecified leprosy decreased over the years (*p* = 0.001), while the diagnosis of lepromatous leprosy increased (*p* = 0.001).

In terms of comorbidities, there was an increase in arterial hypertension (*p* < 0.001), dyslipidemia (*p* < 0.001), and acute kidney failure (*p* < 0.001), with no significant differences in the rest of the comorbidities evaluated. Another important aspect was the slight increase in cases of leprosy with hepatitis B virus co-infection (*p* = 0.02). On the other hand, the frequency of leprosy complications remained constant over the years, with the exception of the increase in cases of neuropathy (*p* = 0.013) (Table 4).

The number of cases increased in the Valencian Community (*p* < 0.001) and Madrid (*p* = 0.005), while they decreased in Andalusia (*p* = 0.001), the Canary Islands (*p*= 0.001), Catalonia (*p* = 0.007) and Galicia (*p* = 0.024) (Figure 2). 

## 4. Discussion

Admissions for leprosy in Spain have decreased over the years, including in the autonomous communities with the highest prevalence. The population of patients hospitalized for this disease has become older, with more comorbidities, in consonance with the overall trends in admissions for other chronic diseases [21]. The distribution of cases between sexes has also changed, with a gradual shift towards more women being admitted for leprosy. This could be related to the longer life expectancy in women [22] or the increase in the female immigrant population [7]. Length of stay has decreased over the years, as also observed in admissions for most other diseases as hepatitis or HIV in Spain [21,23].

Admissions for leprosy in Spain have decreased over the years. During 2020, they were deeply affected by the SARS-CoV-2 pandemic. However, in 2021, they have slightly increased (the second year of the SARS-CoV-2 pandemic). Various diseases have experienced decreased reporting at a national level [24,25,26,27]. For example, there has been a decrease in new HIV cases in Italy and Spain [24,25]. In Germany, from week 1–2016 to week 32–2020, there was a drastic decrease in notifications for respiratory diseases (from 86% [measles] to 12% [tuberculosis]), gastrointestinal diseases (from −83% [rotavirus] to −7% [yersiniosis]), and imported vector-borne diseases (−75% [dengue fever] or −73% [malaria]). There has also been a decrease in sexually transmitted diseases (from −28% [hepatitis B] to −12% [syphilis]) [26]. Additionally, in Spain, influenza-related hospitalizations decreased significantly from 36,033 in the 2018–2019 season to 24,764 in the 2019–2020 season, and further to 231 in the 2020–2021 season [27]. 

The frequency of reported cases in the autonomous communities of Valencia and Madrid rose over time. In the Valencian Community, these data are not attributable to a true increase in cases, but rather to local epidemiological and administrative circumstances. Specifically, the hospital discharge records database collects information only from public centers, but in the Valencian Community there is a center for monitoring and admission of patients with leprosy, the Sanatorium San Francisco De Borja Fontiles. Until 2014, the center was run by non-governmental organizations, and thereafter it was subsidized as a medium- and long-stay hospital. The subsequent integration of that center’s data into the public hospital discharge records database artificially increased the number of reported cases in this community.

In Madrid, the increase in cases can be attributed to the rise in immigration in this region, especially from Latin America; indeed, the most frequent country of origin in our cohort, apart from Spain, was Paraguay. These results are in line with the data collected by the State Leprosy Registry as well as the results of other similar studies [28,29], which show that most incident cases are patients from other countries [30,31]. 

The detection of new cases in children indicates the persistence of leprosy transmission within the community [28]. In our study, the low number of admissions in patients under 15 years of age (0.3%) supports the idea that transmission within the country is low, with the majority of leprosy cases admitted being either imported or autochthonous cases presenting with sequelae requiring hospitalization. In contrast, in other autonomous communities where leprosy was classically more prevalent, such as Andalusia, the Canary Islands and Galicia [32], the number of cases has decreased significantly.

This study describes the comorbidities in patients hospitalized with leprosy. The main one was hypertension, present in 21.3% of cases, followed by diabetes mellitus (18.2%). The prevalence of hypertension increased slightly but significantly over the five study periods, while the prevalence of diabetes remained steady. Plantar ulcers often take months or years to heal, affecting quality of life in people with leprosy. The size of the ulcer and its resolution are associated with several factors, such as hypertension, diabetes mellitus, low body mass index, and smoking [16,33,34]. In our study, about 13% of admitted patients were chronic leprosy cases and older people with chronic ulcers, cellulitis, or acute/chronic osteomyelitis, which helps explain the prevalence of hypertension and diabetes mellitus.

Moreover, diabetes and hypertension can cause neuropathy disturbances in leprosy patients [34]. Our study showed a slight increase in neuropathy over the five study periods, which may be related to the steady increase in the prevalence of hypertension and lepromatous leprosy.

Anemia was more prevalent in patients with leprosy compared to the general population [35]. For example, in Nigeria, the prevalence was 53.3% compared with a control group (10.0%) [35]. In our study, the prevalence of anemia was 11.2%, but we didn’t have a control group. We did not observe an increase or decrease in the prevalence of anemia during the different periods of the study. 

The dyslipidemia was present in 10.6% of patients and prevalence remained steady over the five study periods. The literature has not extensively studied hypercholesterolemia and hypertriglyceridemia in patients with leprosy. 

The prevalence rates of hepatitis B and hepatitis C in leprosy patients are higher than in the general population [36]. In our study, the prevalence of hepatitis B and hepatitis C was 2.5% and 4.1%, respectively, while in Spain, the prevalence of hepatitis C was 0.2% and hepatitis B was 0.1% [37,38] and vaccination was implemented in 1994 for all newborns. Each prevalence was lower than in leprosy patients. A recent meta-analysis also found that these co-infections were associated with higher rates of leprosy reactions [36]; unfortunately, we were unable to make this comparison in our study. Additionally, we observed a slight increase in hepatitis B co-infections and neuropathy in patients with leprosy. It is well known that hepatitis B infections can cause polyneuropathy [39], which may explain the increase in polyneuropathy in patients with hepatitis B co-infection.

Among the strengths of this study, it provides a broad vision of the admission trends for patients with leprosy in Spain over 25 years and illustrates the declining impact of this disease. That said, this series does not reflect the number of new cases of leprosy, but rather the number of hospitalizations, two measures which do not necessarily go hand in hand, given that patients with leprosy, even if treated, remain diagnosed with the disease and some of them are readmitted due to leprosy-related neuro-osteo-cutaneous complications or other age-related comorbidities.

This study also has some limitations. It included only patients who were admitted, and each admission—and readmission—was considered a new episode, so some patients will have been included more than once. The information available for each patient is the set of diagnoses coded at hospital discharge, and it is well known that not all hospitalized patients are coded with all diagnoses at discharge, but on some occasions only those considered most relevant by the attending clinician; furthermore, coded diagnoses are not always used. In the particular case of our study, there is also the previously described limitation related to the data from the Sanatorium San Francisco De Borja Fontilles in the Valencian Community, which accounts for a high proportion of patients. Its inclusion only from 2014 onwards limits the interpretation of the data in this region. This center manages the care of patients diagnosed with leprosy many years ago, and this may have contributed to an increase in comorbidities found in this study. Another minor limitation was that data on patients’ nationality were not available before 2016, which limits a comprehensive analysis of the involvement of immigration in leprosy admissions. Moreover, the information regarding the hepatitis B vaccination status of the admitted patients was not available. 

## 5. Conclusions

Admissions for patients with leprosy have decreased over the years. The case numbers likely represent an overestimation of the number of patients, who may be readmitted throughout their lives for different reasons. The age of patients admitted for leprosy, as well as some age-related comorbidities, has increased over time, while the mean length of stay has decreased, and there has been no change in in-hospital mortality or leprosy-associated complications. Admissions have decreased in regions with historically high levels, such as Andalusia, the Canary Islands and Galicia, but in Madrid they have increased, most likely due to immigration.

## Figures and Tables

**Figure 1 life-14-00586-f001:**
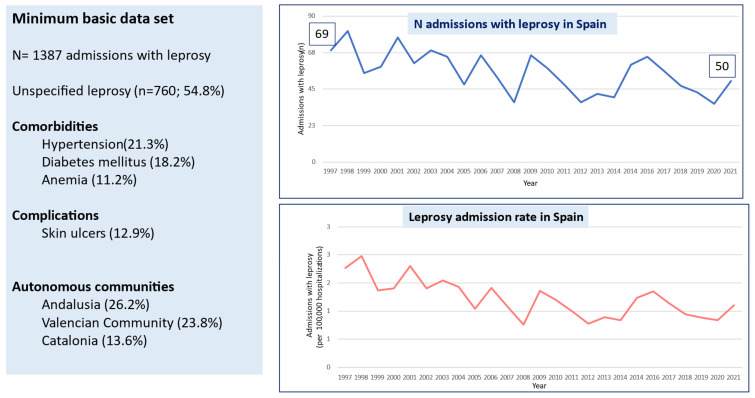
Comorbidities and complications in people admitted with leprosy in Spain, 1997–2021.

**Figure 2 life-14-00586-f002:**
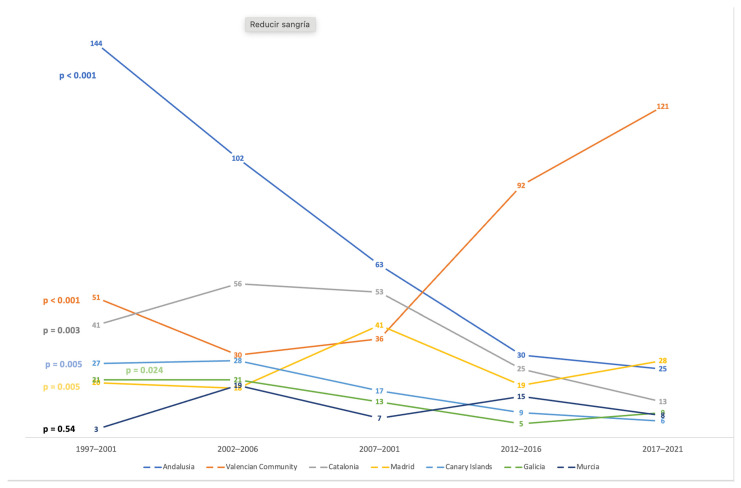
Trends in leprosy admissions in Spain, by autonomous community (Andalusia, Valencian Community, Catalonia, Madrid, Canary Islands, Galicia and Murcia), 1997–2021.

**Table 1 life-14-00586-t001:** Epidemiological characteristics of patients admitted for leprosy, 1997–2021 (N = 1387).

Variables		N (%) *
Age	Age, median (IQR)	68 (57–77)
	<15 years	4 (0.3)
	15–64 years	543 (39.1)
	≥65 years	840 (60.6)
Sex	Male	807 (58.2)
	Female	580 (41.8)
Hospital outcomes	Median (IQR) length of stay, days	8 (4–16)
	Death	83 (6.0)
Diagnosis of leprosy on discharge	Unspecified leprosy	760 (54.8)
	Lepromatous leprosy	428 (30.9)
	Tuberculoid leprosy	80 (5.8)
	Other forms of leprosy	29 (2.1)
	Doubtful leprosy	28 (2)
	Indeterminate leprosy	9 (0.6)
	Sequelae of leprosy †	54 (3.9)
Toxic habits	Tobacco use	84 (6.1)
	Alcohol intake	19 (1.4)
Comorbidities	Arterial hypertension	296 (21.3)
	Diabetes mellitus	253 (18.2)
	Anemia	156 (11.2)
	Dyslipidemia	147 (10.6)
	Chronic kidney failure	110 (7.9)
	Arrhythmia ‡	102 (7.4)
	Heart failure	100 (7.2)
	Chronic obstructive pulmonary disease	79 (5.7)
	Obesity and overweight	65 (4.7)
	Acute renal failure	45 (3.2)
Co-infections	Hepatitis C	57 (4.1)
	HIV	39 (2.8)
	Hepatitis B	34 (2.5)
Leprosy complications	Chronic ulcer	179 (12.9)
	Inflammatory and toxic neuropathy	89 (6.4)
	Cellulitis	69 (5.0)
	Acute and chronic osteomyelitis	65 (4.7)
Microorganism responsible for super-infection	*Staphylococcus*	53 (3.8)
	*Pseudomonas*	34 (2.5)
	*Escherichia coli*	22 (1.6)
	*Streptococcus*	19 (1.4)

IQR: interquartile range; * Unless otherwise noted; † Available from 2016; ‡ Including atrial fibrillation.

**Table 2 life-14-00586-t002:** Patient nationality among admissions for leprosy in Spain, 2016–2021 (N = 297).

Country	N (%)
Spain	235 (79.1)
Paraguay	13 (4.4)
Colombia	7 (2.4)
Senegal	7 (2.4)
Bolivia	3 (2.1)
Brazil	3 (2.1)
Equatorial Guinea	2 (0.7)
Other *	5 (1.7)
Unknown	22 (7.4)

* 1 case each from Cuba, Mali, Mauritania, Nigeria, and the Philippines.

**Table 3 life-14-00586-t003:** Admissions for leprosy in Spain, according to autonomous community, 1997–2021 (N = 1387).

Autonomous Community	N (%)
Andalusia	364 (26.2)
Aragon	9 (0.6)
Asturias	7 (0.5)
Balearic Islands	22 (1.6)
Basque Country	34 (2.5)
Canary Islands	87 (6.3)
Cantabria	8 (0.6)
Castilla y León	11 (0.8)
Castilla-La Mancha	48 (3.5)
Catalonia	188 (13.6)
Ceuta	2 (0.1)
Extremadura	12 (0.9)
Galicia	69 (5)
La Rioja	2 (0.1)
Madrid	126 (9.1)
Murcia	52 (3.7)
Navarra	16 (1.2)
Valencian Community	330 (23.8)

**Table 4 life-14-00586-t004:** Epidemiological and clinical trends in leprosy admissions in Spain, 1997–2021.

Variables		1997–2001 N (%) *	2002–2006 N (%) *	2007–2001 N (%) *	2012–2016 N (%) *	2017–2021 N (%) *	*p*
Age	Age, median (IQR)	65 (57–72)	68 (57–75)	67 (40–76)	71 (54–79)	76 (65–83)	<0.001
	<15 years	2 (0.6)	1 (0.3)	0 (0)	1 (0.4)	0 (0)	<0.001
	15–64 years	156 (45.7)	124 (40.1)	116 (44.4)	89 (36.5)	58 (25)	
	≥65 years	183 (53.7)	184 (59.5)	145 (55.6)	154 (63.1)	174 (75)	
Sex	Male	225 (66)	178 (57.6)	154 (59)	140 (57.4)	110 (47.4)	<0.001
	Female	116 (34)	131 (42.4)	107 (41)	104 (42.6)	122 (52.6)	
Hospital outcomes	Median (IQR) length of stay, days	11 (5–18)	8 (4–16)	8 (4–15)	7 (4–14)	7 (4–15)	0.001
	Death	21 (6.2)	13 (4.2)	12 (4.6)	19 (7.8)	18 (7.8)	0.25
Leprosy diagnosis at discharge	Unspecified leprosy	212 (62.2)	211 (68.3)	159 (60.9)	128 (52.5)	50 (21.6)	0.001
	Lepromatous	89 (26.1)	71 (23)	75 (28.7)	87 (35.7)	106 (45.7)	0.001
	Tuberculoid	30 (8.8)	12 (3.9)	16 (6.1)	13 (5.3)	9 (3.9)	0.045
	Indeterminate	2 (0.6)	3 (1)	2 (0.8)	2 (0.8)	0 (0)	0.46
	Doubtful leprosy	5 (1.5)	5 (1.6)	1 (0.4)	0 (0)	17 (7.3)	0.001
	Other	4 (1.2)	7 (2.3)	8 (3.1)	4 (1.6)	6 (2.6)	0.36
Toxic habits	Tobacco use	22 (6.5)	28 (9.1)	11 (4.2)	21 (8.6)	22 (0.9)	0.018
	Alcohol intake	11 (3.2)	2 (0.6)	5 (1.9)	1 (0.4)	0 (0)	0.002
Comorbidities	Hypertension	51 (15)	63 (20.4)	58 (22.2)	60 (24.6)	64 (27.6)	<0.001
	Diabetes mellitus	55 (16.1)	78 (25.2)	43 (16.5)	41 (16.8)	36 (15.5)	0.25
	Anemia	33 (9.7)	30 (9.7)	41 (15.7)	33 (13.5)	19 (8.2)	0.68
	Dyslipidemia	12 (3.5)	32 (10.4)	28 (10.7)	30 (12.3)	45 (19.4)	<0.001
	Chronic obstructive pulmonary disease	20 (5.9)	22 (7.1)	7 (2.7)	11 (4.5)	19 (8.2)	0.80
	Heart failure	19 (5.6)	30 (9.7)	22 (8.4)	9 (3.7)	20 (8.6)	0.92
	Chronic kidney failure	18 (5.3)	28 (9.1)	27 (10.3)	19 (7.8)	18 (7.9)	0.35
	Obesity and overweight	17 (5)	19 (6.1)	12 (4.6)	4 (1.6)	13 (5.6)	0.38
	Arrhythmia †	13 (3.8)	27 (8.7)	26 (10)	15 (6.1)	21 (9.1)	0.072
	Acute kidney failure	3 (0.9)	7 (2.3)	9 (3.4)	13 (5.3)	13 (5.6)	<0.001
Co-infections	HIV	8 (2.3)	6 (1.9)	7 (2.7)	13 (5.3)	5 (2.2)	0.31
	Hepatitis C	12 (3.5)	9 (2.9)	12 (4.6)	15 (6.1)	9 (3.9)	0.27
	Hepatitis B	2 (0.6)	8 (2.6)	6 (2.3)	12 (4.9)	6 (2.6)	0.02
Complications	Chronic ulcer	49 (14.4)	42 (13.6)	37 (14.2)	32 (13.1)	19 (8.2)	0.059
	Cellulitis	18 (5.3)	15 (4.9)	16 (6.1)	13 (5.3)	7 (3)	0.39
	Acute and chronic osteomyelitis	17 (5)	7 (2.3)	11 (4.2)	14 (5.7)	16 (6.9)	0.10
	Inflammatory and toxic neuropathy	16 (4.7)	15 (4.9)	17 (6.5)	21 (8.6)	20 (8.6)	0.013

IQR: interquartile range; * Unless otherwise noted; † Including atrial fibrillation.

## Data Availability

The datasets generated during the current study are not publicly available due to privacy or ethical restrictions but are available from the corresponding author on reasonable request.

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
