# Peer review of "Comorbidities and Complications in People Admitted for Leprosy in Spain, 1997–2021"

_life, 2024, doi:10.3390/life14050586_

Round 1

Reviewer 1 Report

Comments and Suggestions for Authors

Title of the article: 

Comorbidities and complications in people admitted for leprosy in Spain, 1997-2021

 Manuscript ID:

life-2940991

 General Comments

Thank you for the opportunity to review your interesting manuscript. I found it to be a compelling read, and I have a few considerations to address. In accordance with MDPI Guidelines, I suggest reorganizing the abstract to synthesize the most relevant information into a single paragraph of approximately 200 words (see Major Compulsory Revisions). Additionally, I recommend expanding the discussion to include the potential effects of the SARS-CoV-2 pandemic during the period of 2020-2021 on leprosy diagnosis (see Major Compulsory Revisions). The English syntax is of high quality. All things considered, I recommend acceptance of this manuscript pending major revisions.

 Major Compulsory Revisions

Please consider reorganizing the abstract into a single paragraph of approximately 200 words. According to MDPI Guidelines (https://www.mdpi.com/authors/layout), "the abstract contains a summary of the entire paper and can be up to 200 words long with only one paragraph."

 Your manuscript spans a significant period of time (over 20 years). However, the last two years considered have been deeply affected by the SARS-CoV-2 pandemic. Various diseases have experienced decreased reporting at a national level (for instance, see the decrease in HIV cases in Italy [Quiros-Roldan E, et al. J Public Health Res. 2021. PMID: 34558252]). Therefore, I suggest expanding the discussion to include the potential impact of the pandemic on leprosy diagnosis and comparing it to the available literature.

Statistical analysis

This is a statistically rigorous work.

Author Response

Dear Sr. 

Thank you for your suggestions. I am sending you a point-by-point response

Sincerely

I. Belinchon

Reviewer 2 Report

Comments and Suggestions for Authors

Introduction

Page 1 Line 36-37: Update the prevalence data to reflect the most recent statistics from the World Health Organization (WHO), ensuring the information is current and accurate.

Page 1, Lines 43-45: References 7 and 8 are not in English and therefore cannot be assessed for relevance or accuracy. It is recommended that English translations or equivalent sources be sought for verification.

General comment: It is advisable to elaborate on comorbidities associated with the condition and to discuss the evolution of cases over time. Such an analysis should include trends in prevalence, notable outbreaks, and shifts in demographic profiles of affected populations, to provide a comprehensive understanding of the condition's impact.

Materials and Methods

Page 2, Line 62: Specify the type of study conducted, such as cross-sectional, cohort, case-control, or longitudinal, to give readers a clear understanding of the research design.

General comments:

·      Indicate if there were any exclusion criteria for study participation to clarify the selection process of the subjects.

·      While the methodology section outlines the statistical tests employed, it lacks detail on the levels of statistical significance (e.g., p-value thresholds) considered for various analyses. Incorporating this information would reinforce the findings' credibility and help readers grasp the study's statistical robustness.

Results

General comments: The mention of patients' nationality from 2016 onwards is intriguing; however, the significance of this information in relation to leprosy transmission and its implications for public health policy in Spain requires further exploration. An expanded discussion on how these trends affect public health strategies and leprosy management would be beneficial.

Discussion

Page 9, Line 181: Support this statement with relevant literature to provide a solid foundation for the claims made, enhancing the discussion's credibility.

General comments: The limitation regarding the exclusive inclusion of data from the Sanatorium San Francisco De Borja Fontilles starting from 2014 should be addressed more thoroughly. Discussing the potential impact of this limitation and any steps taken to mitigate its effects would lend more reliability to the study's conclusions.

Abstract

General comments: While the abstract succinctly outlines the study's objectives, methods, and key findings, it falls short of including specific data on the statistical significance of observed trends, such as the decline in hospital admissions and the increase in median age among patients. Detailing these aspects would augment the abstract's detail and informative value.

Overall comment: In conclusion, incorporating these minor revisions will significantly enhance the manuscript's clarity, depth, and scholarly value.

Author Response

(The authors gave the same response as above.)

Reviewer 3 Report

Comments and Suggestions for Authors

This study is an observational study on leprosy hospitalizations in Spain between 1997 and 2021. The objective is to describe clinical characteristics and some demographic changes over the years in Spain. The findings report a decreasing trend in hospitalizations, with a higher proportion of older individuals and a low number of children under the age of 15 years old. This supports the idea that transmission of leprosy is lower and possibly under control within Spain. However, individuals coming from other countries or regions where leprosy is endemic remain a concern. Please find my comments below.

Please provide an IRB statement for the study regarding ethics. Research studies involving humans are of international concern and should not only meet national regulations but also international standards. It is essential to follow and cite proper ethical guidelines. I suggest authors check this with the journal's ethical regulations. At the very least, a statement on the project identification code, date of approval, and the name of the ethics committee or IRB must be addressed.

In Table 1, the median length of stay in days is given as 9 (4-18). However, in line 124 of the text, it states, "The median hospital stay was 9 days (range 0-6188 days)..." This suggests a hospital stay of 17 years (6188 days), which seems unusually long unless it's comorbid with other problems such as psychiatric issues or coma. In Table 4, the longest hospital stay ever recorded was between 4 and 31 days between 1997 and 2021. These three sets of data are conflicting and need to be checked and corrected for consistency.

Can authors please elaborate more on this statement; "Furthermore, the low number of admissions in patients under 15 years of age (0.3%) supports the idea that transmission within the country is low,.."? How authors make these interpretations from the study?

The age variable given in Table 1 is 68 (57-77), whereas Table 4 shows an average median age of 69.4 with an IQR range between 40 and 83. This discrepancy should be checked for clarity and accuracy. Am I reading it correctly?

Isn't Table 1 showing the total numbers between 1997 and 2021, which is the same as Table 4? If so, why not create two tables and add one more column to the end of Table 4 to show a total?

In the introduction, the impression of leprosy counts being higher and the likelihood of immigration from Brazil is mentioned. However, in the results, it is stated to be 2.1%, ranking fifth in Table 2. Is there any scientific explanation why Brazil specifically has been suspected of the transmission of leprosy in Spain even the results show it`s around 2%?

Author Response

(The authors gave the same response as above.)

Reviewer 4 Report

Comments and Suggestions for Authors

This is an interesting article about the evolution of hospital admissions for leprosy in Spain from 1997 to 2021. The authors demonstrated how admissions of patients for leprosy have decreased over the years and described how clinical and epidemiological characteristics of these patients changed in the last years.

In the following lines I will report some suggestions to improve the manuscript:

1.       I suggest to report the epidemiological and clinical trends of leprosy from 1997 to 2021 in graphic instead of in tables (you may change table 4,5 in graphics). You may indicate the statistical significant results with * putting them in the legend.

2.       I suggest to do some hypotheses in the discussion to explain two results which authors found in their study: the increase of HBV co-infections and of neuropathy in patients with leprosy. Is there an increase of HBV infections in population of Spain and in particular of Madrid and Valencia? Can the increase of neuropathy be associated with the increase of lepromatous leprosy?

Comments on the Quality of English Language

Minor mistakes of English language should be improved.

Author Response

(The authors gave the same response as above.)

Round 2

Reviewer 1 Report

Comments and Suggestions for Authors

All issues have been addressed.

Author Response

Thank you very much for your words

Reviewer 3 Report

Comments and Suggestions for Authors

Thanks for addressing my concerns. I have no more comments. Congrats. 

Author Response

Thank you very much for your words